# A Review of the Impact of Climate Change on the Epidemiology of Gastrointestinal Nematode Infections in Small Ruminants and Wildlife in Tropical Conditions

**DOI:** 10.3390/pathogens11020148

**Published:** 2022-01-24

**Authors:** Carlos Ramón Bautista-Garfias, Gloria Sarahi Castañeda-Ramírez, Zaira Magdalena Estrada-Reyes, Filippe Elias de Freitas Soares, Javier Ventura-Cordero, Pedro Geraldo González-Pech, Erick R. Morgan, Jesús Soria-Ruiz, Guillermo López-Guillén, Liliana Aguilar-Marcelino

**Affiliations:** 1National Center for Disciplinary Research in Animal Health and Safety (INIFAP), Km 11 Federal Road Cuernavaca-Cuautla, Jiutepec 62550, MR, Mexico; bautista.carlos@inifap.gob.mx (C.R.B.-G.); glosahi@gmail.com (G.S.C.-R.); 2National Institute of Research for Forestry Agricultural and Livestock (INIFAP), Experimental Station Rosario Izapa, Tuxtla Chico 30780, CS, Mexico; lopez.guillermo@inifap.gob.mx; 3Department of Animal Science, North Carolina Agricultural and Technical State University, Greensboro, NC 27411, USA; zmestradareyes@ncat.edu; 4Department of Chemistry, Universidade Federal de Lavras, Lavras 37200-000, MG, Brazil; filippeufv@yahoo.com.br; 5School of Biological Sciences, Queen’s University Belfast, Chlorine Gardens, Belfast BT9 5BL, UK; J.Ventura-Cordero@qub.ac.uk (J.V.-C.); Eric.Morgan@qub.ac.uk (E.R.M.); 6Faculty of Veterinary Medicine and Zootechnics, Autonomous University of Yucatán, Km 15.5 Road Mérida-Xmatkuil, Mérida 97100, YU, Mexico; pedro.gonzalez@correo.uady.mx; 7Geomatics Lab, National Institute of Research for Forestry Agricultural and Livestock (INIFAP), Zinacantepec 52107, MX, Mexico; soria.jesus@inifap.gob.mx

**Keywords:** global warming, goat, *Haemonchus*, selection of parasite resistance

## Abstract

Climate change is causing detrimental changes in living organisms, including pathogens. This review aimed to determine how climate change has impacted livestock system management, and consequently, what factors influenced the gastrointestinal nematodes epidemiology in small ruminants under tropical conditions. The latter is orientated to find out the possible solutions responding to climate change adverse effects. Climate factors that affect the patterns of transmission of gastrointestinal parasites of domesticated ruminants are reviewed. Climate change has modified the behavior of several animal species, including parasites. For this reason, new control methods are required for controlling parasitic infections in livestock animals. After a pertinent literature analysis, conclusions and perspectives of control are given.

## 1. Introduction

Climate change (CC) is a modification in the properties of the climate system that persists for several decades or longer. Natural processes, such as changes in the Sun’s radiation, volcanoes or internal variability in the climate system, or changes in the composition of the atmosphere or land use derived from human activity, can contribute to CC. In this context, it has been indicated that CC is making “*El niño*” phenomena (a climate cycle in the Pacific Ocean with a global impact on weather patterns) more intense, with catastrophic consequences [1].

CC has a negative impact on living organisms and can affect the transmission patterns of different pathogens (viruses, bacteria, fungi, parasites) which infect wild and livestock animals, and humans. Modifications in disease transmission due to CC include changes in the rates of development, mortality, and reproduction of free-living stages of parasites, vectors and hosts; behavioral changes in parasites, vectors and hosts; and alterations of host susceptibility by modification of the immune and stress response, and physiology ([Fig pathogens-11-00148-g001]).

These interactions between hosts and pathogens are complex and dynamic [2,3]. For parasitic nematodes, it has been indicated that CC may alter the patterns of development of the free-living stages (*Trichostrongylus* sp. of ungulates) in Nordic countries [4]. In wild animals infected with *Trichinella* species, changes in the distribution area of *Trichinella nativa*, *Trichinella* T6 and *Trichinella britovi* have been suggested [5]. Studies with livestock animals have been conducted to develop mechanistic models that integrate CC, parasite infection and host movements [6].

In ruminants, free living stages of gastrointestinal (GI) nematodes, such as *Haemonchus contortus*, are sensitive to temperature and humidity changes; thus, it is possible that CC could have a negative effect on ruminant infection patterns [7].

Although some predictive models indicate that up to 30% of parasitic worms are threatened by extinction due to CC, it is possible that parasite richness could still increase in some places due to the fact that successful parasitic species can invade temperate ecosystems and replace native species with unpredictable ecological consequences [8]. Morgan and van Dijk et al. [9] suggested that the integration of environmental conditions in predictive models of may help to predict periods of high parasite infection and facilitate the development of deworming practices that target these periods of infection in animal operations. The current review aimed to determine how CC has impacted the small ruminant operations and wildlife, and to identify factors influencing the epidemiology of gastrointestinal nematode infections in small ruminants under tropical conditions based on previous research findings. The latter is orientated to find out possible solutions responding to CC adverse effects.

## 2. Influence of Climate Change on Wild Mammals in Mexico

Wild mammals are receiving less attention than livestock animals in Mexico [10], modeled two scenarios (liberal and conservative) with 416 mammals out of 1870 species (including birds, mammals and butterflies), and they reported a reduction in the number of species in the distributional area. Generally, a reduction in the distributional area is considered as a signal of species extinction. Similarly, analyzed circulation models which evaluated the effects of climate change on 61 mammal species distributed according to five bioclimatic zones in Mexico [11]. Mammal species showed an individual response to the climate change scenarios, and it was suggested that approximately half of these species would reduce 50% of their distribution area, and some species, such as *Romerolagus diazi*, will disappear by 2050.

Furthermore, CC harms the food supply for the human population. A good example of this phenomenon is with rural communities from Mexican wetlands.

The primary food sources for these communities are wild mammals, and reductions in these species cause scarcities in goods and food supplies [12]. Therefore, some rural communities are at higher risk of food scarcity when the weather drastically fluctuates under extreme climatic events [12].

A pivotal program to preserve wild animals, named extensive wildlife management units (eWMU), was released in 1997. Gómez-Aíza et al. [13] reported that 64% of the municipalities in Mexico lost vegetation cover from 2002 to 2011. It might be expected that those wild animals in the eWMU are losing food; therefore, a higher effort is required to maintain animal welfare. The latter was more evident when the municipalities dedicated less than 10% of the area to eWMU [13]. Conversely, some communities have not been affected by climate change, i.e., common vampire bats (*Desmodus rotundus*) from North America using five species distribution models suggested that their expansion towards Mexico–United States borderlands in the next 60 years appeared improbable [14].

An incredible effort is required to understand the impact of CC on wildlife communities in Mexico. Therefore, it is a key chance to design studies involving a multidisciplinary approach (climatic conditions, nutrition, reproduction, survivorship, etc.). Furthermore, that point of view will enable us to develop an overall picture of the current and most crucial further situations of wild animal communities.

Mammals are distributed in different climates and geographical regions, and they play an essential role in ecosystems [15].

In Mexico, most studies with mammals have been carried out with Chiroptera [bats] and Rodentia [rodents]. However, few studies have studied parasitic nematodes in these and other species of mammals [16]. It has been suggested that bats have a great diversity of nematodes due to their habitat diversity and feeding behavior [17]. For rodents, Thominx, *Vexillata vexillata* and *Vexillata dessetae* have been reported. In addition, *Raillietina demerariensis*, *Gongylonema neoplasticum*, *Hymenolepis diminuta* and *Moniliformis moniliformis* have been studied as zoonotic [18]. Rodents have a great diversity of helminths, and some of them can infect humans [19].

In marsupials from Latin America and Mexico, the following nematodes have been reported: *Aspidodera raillieti*, *Didelphosotrongylus hayesi*, *Cruzia tentaculata*, *Gnathostoma turgidum*, Gongylonema sp. Turgida turgida, *Vianna viannai*, travassostrongylus, *Trichuris didelphis* and *Capillariinae gen* sp. [20].

Studies on armadillos are few, but what is known today is of importance because some zoonoses of armadillos have been reported [21]. Some of the reported pathogenic agents of armadillos that can affect humans are: strains of *Salmonella* sp., *Borrelia* sp., *Leptospira* sp., Trypanosomes and helminths. In the case of helminths, there are reports of the presence of *Ascaris lumbricoides* and *Ancylostoma caninum*, which can be transmitted to humans and canines [22].

Order Pilosa: This order includes anteaters, a mammal found mainly in Mexico and South America, which can be found in forests and green areas. As its name suggests, its diet is based on termites and ants. Most of the reported studies are on public health (*Trypanosoma legery*, *T. cruzi*, *T. rangeli* and *Leishmania mexicana*). However, there are few reports on helminths reported in anteaters. In one study, 10 animals were found (24 h) after being run over and killed in Tabasco and Chiapas, Mexico. Subsequently, the different organs were dissected in the laboratory to obtain the parasites. Four species of nematodes (three gastrointestinal and one lung), one cestode, one flea, one tick and one mite were found in these individuals [23]. On the other hand, due to the higher population of anteaters in other countries, the presence of nematodes such as *Trichuris* sp., *Strongyloides* sp., *Ascaris* sp., and *Ancylostoma* sp. have been reported [24].

Order non-human primates: These are considered among the most illegally hunted animals by humans, for consumption or as exotic pets [25]. This order mainly comprises several species of monkeys, gorillas and lemurs. Three species of monkeys (*Alouatta pigra*, *Alouatta palliata mexicana*, *Ateles geoffroyi*) can currently be found in Mexico, and these were isolated in protected areas for conservation. However, before the protected areas, these animals were hunted, and their habitat destroyed. There are several studies and reports of diseases in monkeys in some parts of the world [26]. In the case of nematodes in monkeys in Mexico, the following have been reported: for the species *A. palliata*, the nematodes *Parabronema bonne*, *A. lumbricoides* and *Tripanoxyuris minutus* were identified [27,28]; for the species *A. geoffroyi*, the nematodes *Strongyloides stercolaris* and *Enterobius vermicularis* have been detected [29,30].

In Mexico, there is a lack of studies on the behavior of wildlife nematodes as a result of CC. However, in other countries, it has been pointed out that in biologically diverse nematodes, CC contributes to increases in the range of distribution, colonization of new hosts and modification of their development cycles [31].

## 3. Effect of Climate Change on Nematode Parasites of Livestock

CC has severely impacted the livestock industry. The increase in temperature and atmospheric carbon dioxide (CO_2_) concentration, and precipitation variation has negatively impacted animal performance, and the quality of feed crop and forage, water availability and consumption, disease transmission [32,33], reproduction and biodiversity [33].

Previous studies have modelled the direct and indirect effects of different CC scenarios on pasture production [34], on crop livestock, pastoral systems [35] and helminths of livestock [36], and have reported production losses from 30% to 60% [37]. Consequently, CC could cause a redistribution of the global animal inventory, where reductions in cattle production could promote an increase in small ruminant operations. Among the small ruminant species, goats are generally considered as more tolerant to warmer environments than sheep and other domesticated ruminant species [38,39]. The exceptional phenotypic characteristics of goats allow them to use minimum energy to maintain a constant core temperature in temperate conditions (from 25 to 30 °C). Additionally, the unique ability of goats to desiccate feces, concentrate urine, reduce evaporative water loss, and use rumen as a water reservoir [40] gives them survival advantages when compared with sheep. However, additional energy is required for thermal adaptation to ambient temperatures above 30 °C, which causes impairment in productive performance and harmful consequences for animal health, product quality and reproduction [38,41]. In this context, goat production represents an option to overcome elevated environmental temperatures and reduce production losses derived from CC. It is possible that goats will spread their distribution into agroecological zones with no direct competition with agriculture lands and cattle farms will be forced to strong reorientate to more thermotolerant systems.

Redistribution of the ruminant inventory will rely on the ability of the production system to overcome the negative effects derived from CC and the availability of feeding resources.

One noticeable indirect effect of changing temperatures and rainfall patterns on parasite transmission potential is through grazing season length, with more extended grazing periods in temperate areas, which instigates the accumulation of infective parasite larvae on pastures, and subsequent animal infection [42]. In addition, an increase in rainfall precipitation could promote a suitable environment for the development of water-dependent parasites such as liver flukes.

Furthermore, current intensive livestock production systems, with increased housing and feeding, decrease grass-transmitted parasites such as gastrointestinal nematodes, but promote coccidia infections and other diseases associated with poor hygiene.

At the same time, food–feed competition in areas suitable for cultivation, along with increasing demand for animal-derived products and consumer preferences, dictate that a market for extensively grazed livestock production will remain, and is often the only viable system for resource-poor smallholder farmers [43]. Livestock management systems are shaped by climate, and dictate not only parasite epidemiology, but also the practicality of novel and alternative control tools, many of which are difficult to deliver in extensive systems.

The impact of parasites, and hence the benefits of control, depend on the economic and social contexts of livestock production [44]. The effects of CC on farm systems through adaptation, mitigation and changing policy and market forces are therefore at least as important to outcomes as the direct effects on host or parasite biology. Due to the huge uncertainty around CC projections and its effects on complex systems, and the likelihood of increased weather variability, it is essential to support the capacity to adapt. The erosion of genetic diversity in livestock is a concern, especially in breeds suited for extensive and low-input production, because this is a key component of adaptive capacity for parasite management as well as other animal health constraints.

The global problem of gastrointestinal nematodes and parasitic resistance in domestic ruminants remains one of the main obstacles to their husbandry [31,45,46,47]. In this context, Ref. [47] stated that the internal parasitism of ruminants is one of the main obstacles to animal production in pasture systems in tropical and subtropical regions. In some regions, verminosis is the leading of death in small ruminants, especially in young animals. These parasitic nematodes (GIN) are popularly known as strongylides and are present year-round in domestic ruminants’ pastures.

The literature reports a series of studies that depict this problem. However, in general, most of them emphasize poor sanitary management, negligence using anthelmintic drugs on a large scale, and climatic, meteorological and ecological conditions as predisposing factors to the prevalence of helminth infections [48]. In this line of reasoning, the importance of recognizing the problems caused by GINs around the world is evident. Parasitic control should, above all, be based on the knowledge of population dynamics and the epidemiology of GINs, their characteristics, and the main species present in the most diverse types of ruminant breeding worldwide.

In tropical countries, the genera *Trichostrongylus* sp., *Cooperia* sp., *Oesophagostomum* sp., *Haemonchus* sp., have well-known prevalence rates. They are the main GINs found in the routine of breeding domestic ruminants, causing numerous problems to animal health. Reports show that sheep and goats can be infected by the same GIN species, mainly the Trichostrongylidae family. However, a recent study by Costa-Junior et al. [49] suggested that the effects of nematodes in small ruminants have specific immunological and physiological differences. Those authors also emphasized that understanding these factors is extremely important for developing sustainable strategies for parasite control and delaying the appearance of parasite resistance.

Thus, knowledge of the contamination degree of pastures by pre-parasitic forms of GIN (L_3_) is of importance for epidemiology. It can help us to determine the risk of infection of animals, and integrated control programs can be generated with the data obtained.

Kaplan et al. [50] also indicate that studies carried out in the last decade in Canada, Europe, Australia, and Brazil reported a high prevalence of resistance to several anthelmintics. These authors also provide an excellent description of the prevalence of parasitic resistance in small ruminants versus cattle, mentioning that the problem is more serious in sheep and goats. Navarre et al. [51] reported an increase in anthelmintic resistance, and new strategies must be developed, promoting the gradual decrease in anthelmintic use. Furthermore, the maintenance of a good integrated management system for the control of GIN (health and pastures) is emphasized. In another aspect, the phenomenon of “refugia” (which is preserving a proportion of the parasite population not exposed to treatment) has been proposed as a measure that can be incorporated into an integrated parasite control system [52].

According to Amarante et al. [53], climate and vegetation have a great influence on the development and survival of larvae, which results in seasonal variation in the presence of infective larvae in the pasture throughout the year. In this sense, it is important to highlight the abundance of infective stage forms of GIN (L_3_) in the environment and for long periods in the pastures of domestic ruminants [54,55].

The life cycles of these parasites are influenced by a series of factors, which include climatic variations. Infecting larvae are active, move erratically and apparently migrate in any direction [56], including to vegetation located near feces. The infecting larvae (L_3_) of GINs need humidity to survive in the grasslands; however, when the rains are scarce, the populations of the GIN decrease due to this situation.

Another important point raised by De Almeida et al. is that low rainfall conditions associated with relatively mild temperatures can determine the survival of L_3_ inside fecal pellets for extended periods, representing a source of contamination of pastures [47].

Precipitation moistens and softens the feces, allowing the larvae to leave the feces and migrate to the pasture. The moisture present in the pastures at certain times of day could serve as a kind of “road” to larval movement; thus, the climate is an important factors that deserves attention in parasite control. Bishop and Stear [57] mention that even in periods of a lack of rain and an unhealthy environment for the survival of larval stages, it is possible to find L_3_ that survived. This information is important and should be taken into account to establish an adequate management plan. Thus, a strategic plan for parasite control that recognizes the climatic particularities of regions that produce ruminant herds seems to be an important premise, although is seldom used.

Several studies have shown how climatic changes influence the temporal and spatial distribution of nematodes [58]. One critical factor for nematodes is temperature; when the temperature increases, the cycles are shortened, and the populations of these parasites increase. This has mainly been observed in tropical areas. In warm tropical areas, infective larvae of *H. contortus* can be obtained from fecal cultures within four to five days after hatching, whereas in temperate laboratory conditions, it can take seven to ten days [59,60].

Differences between the development of *H. contortus* have already been reported over the years, showing differences between warm (accelerated) and cold (delayed) climates [61]. In addition, it has already been shown that nematodes from different geographical areas exhibit variability in in vitro tests [62,63].

Another factor influencing nematode survival in sub-humid climates is after the rainy season [64]. Several authors agree that the highest numbers of parasites occur in the post-rainy season. In a study on calves, a lower number of nematodes was observed due to the lack of moisture before the rains, showing low numbers in the paddocks [65].

## 4. Selection for Parasite Resistance in Small Ruminants under Climate Change

The selection of animals for parasite resistance could be a sustainable response to the increased risk from parasites due to climate change. Therefore, it has been proposed as one of the most promising, natural, sustainable, and affordable alternatives to synthetic drugs because it adds permanent genetic changes, can be transmitted to future generations, improves animal performance, reduces the infectivity of grazing pastures (reduces the number of infective larvae in pastures), and enhances the socio-economic viability of ruminant producers. The main benefit of genetically parasite-resistant animals is the epidemiological effect derived from reduced FEC [57]. Animals that have been selected for parasite resistance excrete fewer eggs in the feces [66,67], which leads to reduced pasture larval contamination and greater performance. Consequently, animals are exposed to a lower burden of infective larvae which benefits all animals grazing the same pasture, including susceptible individuals. Thus, the use of genetically parasite-resistant individuals promotes stewardship by enhancing the quality of the pastures (land) and reducing contamination of the soil and water.

### 4.1. Selection for Parasite Resistance, Resilience, or Tolerance

Selection for parasite resistance, rather than resilience or tolerance, has been an issue often debated by the scientific community. Although resilience is the ability of an animal to maintain performance and tolerance is the ability to maintain homeostasis in the presence of GIN infections [68], selection for these phenotypes is complex to implement under practical conditions [69]. Furthermore, tolerance and resilience increase infectivity of the pastures, which can be major problem for ewes/does that pasture immediately after lambing/kidding, and for susceptible lambs/kids and adult individuals.

Additionally, the determination of resilient and tolerant phenotypes is technically challenging because animal performance must be measurable for different infection levels and requires accurate data. In contrast, parasite-resistant sheep utilize innate and acquire immune responses to limit the establishment and fecundity of the parasite. Furthermore, selection for parasite resistance reduces the number of eggs excreted into the environment, resulting in the reduced utilization of anthelmintics and decreased anthelmintic resistance, which benefits producers and the environment.

### 4.2. Parasite Resistance: A Polygenic Trait with Small Effects

Genetic variation for resistance to GIN has been demonstrated in several breeds of small ruminants selected based on FEC performance [70]. Previous studies have reported that heritability for FEC is highly variable, and goat breeds have lower heritability estimates (0.1–0.33) when compared with sheep breeds (0.01–0.4). Among sheep breeds, Merino, German Merino, Red Maasai, and Dorset–Rambouillet–Finn have higher heritability estimates (0.35–0.39) than Dorper, Soay and Scottish Blackface sheep (0.10–0.18) [71,72].

For goats, higher heritability estimates (0.2–0.3) have been observed in crossbred cashmere goats when compared with Creole (0.10–0.14) and Galla goats (0.13) [73]. Estimations of FEC heritability require accurate measurements, and under natural infections and grazing systems, animals are exposed to different levels of infection, which can be a limitation for FEC assessment [74].

Consequently, FEC phenotyping should be accompanied by other phenotypic measures such as FAMACHA score, percentage cell volume (PCV), immune response (eosinophil and CD4+ lymphocyte counts, and immunoglobulin concentrations) and production traits (average daily gain: ADG, residual feed intake) to evaluate parasite resistance in small ruminants [74]. From these measures, FEC and PVC scores have been widely utilized by producers for assessments of parasite resistance, and have facilitated the accumulation of records [73]. FEC and FAMACHA score recording could be implemented by producers to select parasite-resistant sires and dams using ranked predicted breeding values to improve parasite resistance, with the goal of having sheep that are able to perform better under environments with relatively severe internal parasite infections. With the advancement of next-generation sequencing and lower costs of genotyping, the livestock industry has adopted new genetic technologies to improve production progress across the globe [75].

Genomic regions associated with other complex traits, such as hypertrophy in skeletal muscles (callipyge), have been successfully identified in sheep; however, the genetic architecture underlying parasite resistance is poorly understood, and previous studies suggest that many genes with small effects contribute to parasite resistance [76]. Furthermore, a common conclusion across these studies is that each breed represents a unique genomic landscape, and a DNA marker associated with parasite resistance in a specific breed may not have the same effect in another breed [77].

One of the early methods utilized for the identification of genetic markers controlling parasite resistance was candidate gene analysis. In this approach, variations (microsatellites, single nucleotide polymorphisms, etc.) in genes related to the immune response against gastrointestinal nematodes [78] and the interferon gamma (*INF-**γ*) gene and the major histocompatibility complex (MHC) region are mainly associated with FEC [79,80,81,82,83]. This approach includes preselected candidate genes distributed across the genome and is effective in highlighting chromosomal regions contributing to parasite resistance in natural or artificial infections with *H. contortus* and other GINs. However, as discussed by [77], the major concerns on candidate analysis are the low statistical power and the validation procedures required to confirm functional candidate genes.

After experimentation with candidate genes, quantitative trait loci (QTL) analysis and genome-wide association studies (GWAS) emerged as powerful methodologies to elucidate potential genomic regions associated with parasite resistance in small ruminants. In QTL studies, microsatellite markers (typically from 100 to 200 markers) have been utilized to localize causative variants (controlling parasite resistance) in pedigreed populations (with known sire and dam information) infected with *H. contortus* [84] or other GINs [70,85,86,87,88]. These studies have identified QTLs for FEC in chromosomes 1, 2, 3, 4, 5, 6, 7, 8, 10, 12, 13, 14, 20, 21, 22, 23 and 26.

For GWAS, single nucleotide polymorphisms (SNPs) have been widely used due to their abundance, distribution across the genome and low cost of genotyping. The majority of GWAS for parasite resistance have utilized medium-coverage SNP chips that enable the study of genetic variation of more than 50,000 SNPs per individual. A wide range of SNP markers associated with parasite resistance has been identified in OAR 1, 2, 3, 6, 8, 10, 11, 12, 13 and 21 using genome-wide scans and SNP data [76]. Several of these loci associated with parasite resistance have been identified within or close to immune-response-related genes. Despite the success of GWAS to identify potential loci associated with parasite resistance, no major genes with large effects have been located, and the causal variants have not been elucidated.

It is possible that some of the significant loci providing the strongest association signals may tag other co-inherited variants. For this reason, the confirmation of GWAS results is necessary to validate potential genetic markers controlling parasite resistance in small ruminants. Additionally, other authors have suggested that the mechanisms used for controlling *H. contortus* infection vary between sheep and goat breeds [76]; consequently, it is possible that the genetic markers controlling this complex trait in sheep are not utilized the same for goats. Another important aspect which these studies must consider after the completion of genetic marker validation is the study of the functionality of these genetic variants [89].

### 4.3. Effects of Selection on Other Production Traits

An important aspect to be considered during selection for parasite resistance is the impact on other traits; for this reason, the selection program should evaluate key production traits associated with profitability (such as wool and meat production) in small ruminants. Several authors have reported that selection for parasite resistance in Perendale, Romney and Australian Merino sheep results in unfavorable responses for production traits (wool traits) [90,91,92]. Consequently, frequent assessment of the effects of selection for parasite resistance should be performed to mitigate the negative impact on production traits.

## 5. Future Studies Aiming to Enhance the Adaptability of Small Ruminants to Parasite Infections in a Changing Climate

It is suggested to design future studies in which it would be desirable to evaluate different substances previously assessed in other animals; for example, edible mushrooms and environmentally friendly Lactobacilli to induce natural immunity and enhance adaptive immunity in small ruminants, capable of protecting them from parasite infections in a changing climate. It is also suggested to try food for ruminants containing indigenous plants with natural components which affect GIN parasites. Similarly, it would be desirable to carry out experiments in which an integrative parasite control without the use of chemical parasiticides is an important factor. It is understood that raising small ruminants should include improved management practices.

## 6. Conclusions

Briefly, climate change has had harmful effects on small ruminants in tropical areas, such as heat stress, limited and low-quality pasture availability, and changes in the epidemiology of GIN, which requires new studies on the biology of these to establish new control measures. The information analyzed in this review indicates that: (1) climate change is altering the behavior of animals, including feral animals, ruminants, and GIN of the latter; (2) selection for genetic parasite resistance has been proposed as one of the most promising natural, sustainable, and affordable alternatives to synthetic drugs because it adds permanent genetic changes, can be transmitted to future generations, improves animal performance, reduces the infectivity of grazing pastures and enhances the socio-economic viability of ruminant producers.

## Figures and Tables

**Figure 1 pathogens-11-00148-g001:**
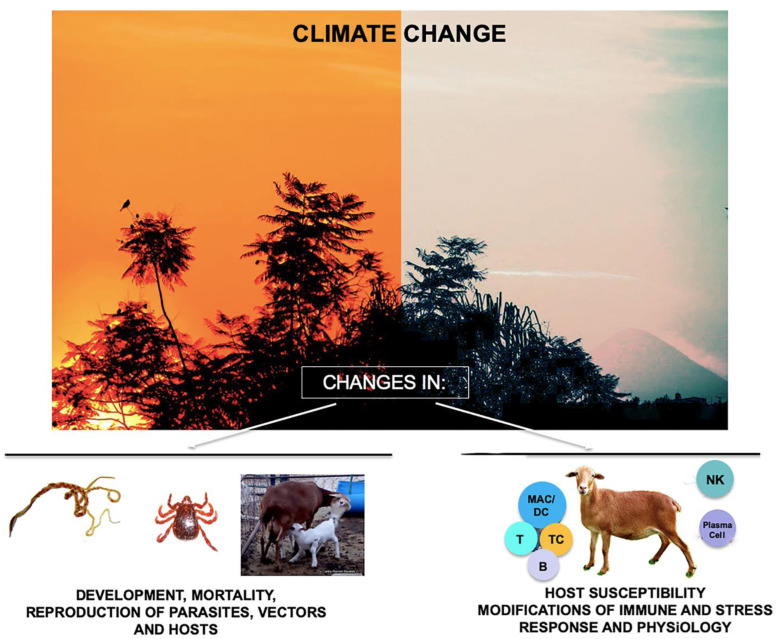
On the left side are shown a *Haemonchus contortus* male and female copulating, then a *Rhipicephalus* sp. tick and a sheep feeding her lamb. On the right side, the immune cells of a sheep are shown: MAC/DC: macrophage/dendritic cell, T:T cell; CT: cytotoxic T cell; B:B cell; NK: natural killer cell; and plasma cell (Figure elaborated by CR Bautista-Garfias).

## Data Availability

The data that support the findings of this study are available from the corresponding author, upon reasonable request.

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
