# Peer review of "A Review of the Impact of Climate Change on the Epidemiology of Gastrointestinal Nematode Infections in Small Ruminants and Wildlife in Tropical Conditions"

_pathogens, 2022, doi:10.3390/pathogens11020148_

Round 1

Reviewer 1 Report

See attached file for detailed comments to incorporate 

Author Response

Reviewer 1

Title of the paper: A Review of the Impact of Climate Change on the Epidemiology of Gastrointestinal Nematode Infections in Ruminants and Wildlife General comments: The review is well written and addressing much needed issue of the time. However, following comments may be considered to enhance the significance of review.

Authors answers: Comments are greatly appreciated.

  1. The data on the epidemiology of the gastrointestinal parasites may be given and a brief trend of increase or decline with respect to climate change may be added.

Authors answers: The correction was made.

  1. There is a continuity in the information given, However, authors major focus is on the impact of climate change on animals rather than gastrointestinal parasites.

Authors answers: The correction was made. Page 2.

  1. More upto date and relevant literature must be cited. Some relevant papers i.e. Okulewicz, 2017; Rose et al., 2014, Morgan and Dijik, 2012 are given as example which must be cited but are missing in the present review.

Authors answers: The correction was made. Page 2.

  1. The use of abbreviations is not uniform i.e. sometimes full forms are not given before use of abbreviations and most of the time, full forms are given again and again after using abbreviations. So, authors are advised to bring uniformity in use of abbreviations.

Authors answers: The correction was made. Page 1-12.

  1. The format of bibliography is also not uniform. Authors must follow journal guidelines for giving full references.

Authors answers: The correction was made. Page 13-18.

Line No 26. Keeping in view the title of paper, the word “Gastrointestinal” nematodes may be added in Key words.

Authors answers: The correction was made. Page 1.

Line 33. Brief elaboration of “El niño” phenomenon may be given in 1-2 lines.

Authors answers: The correction was made. Page 1.

Line 39. The word “Vectors and Host” is used twice, rewrite sentence to clarify the statement.

Authors answers: The correction was made. Page1.

Line 46. The figure No. and caption is missing. The caption must be self-explanatory and in a sentence form.

Authors answers: The correction was made. Page 1.

Line 52. The word “Gastrointestinal” should be abbreviated as “GI” for use in further text of paper. Similarly, the “Haemonchus contortus” would be abbreviated as H. contortus for further use in text. Overall, repeated names be abbreviated once and may be used later in text.

Authors answers: The correction was made. Page 2.

Line 69 106 132 159 178

Authors answers: The correction was made. Page 2, 3, 4, 5.

Use abbreviation CC

Authors answers: The correction was made. Page 4.

Line 108- 111.  The sentence is too long and ambiguous. Its difficult to understand what authors wants to state. Rewrite/elaborate sentence.

Authors answers: The correction was made. Page 3.

Line 195. moniliformis have been studied due to----??? Incomplete sentence

Authors answers: The correction was made. Page 5.

Reviewer 2 Report

Dear editor,

After reviewing  the Manuscript ID: pathogens-1437894.  A Review of the Impact of Climate Change on the Epidemiology of Gastrointestinal Nematode Infections in Ruminants and Wildlife.

The writing presents a topic of interest that impacts animal production and the economy of producers. State of the art on the interaction between animals and parasitism under the environmental challenge of climate change is essential for science and humanity. The structure of the review is interesting and raises the current situation of the actors and their future. It shows the areas of opportunity that must be addressed to decrease the negative impact of climate change on animal welfare and animal production. It proposes the selection of animals where production and resistance to pathogens are integrated. This document will be of interest to scientists and producers of Small ruminants around the world.

As a result, I recommend publication the manuscript in the journal when the authors have changed the next observations.

1) The title mentions that the topic is about ruminants. However, more than 90% of the text refers to Small ruminants (sheep and goats). Therefore I suggest that this point must be adjusted in the title. If the authors prefer to keep Ruminants, then they must include more informatión about GIN infection in cattle. But ticks are more critical than GIN in this animal.

A Review of the Impact of Climate Change on the Epidemiology of Gastrointestinal Nematode Infections in Small Ruminants and Wildlife.

Line 33. Could you explain “el niño” phenomenon”?

Line 94. Please upload to line 93

Line 116-119. This paragraph needs an edition. Temperature and rainfall patterns influence parasite transmission. However, those factors are directly related to parasite burden.

Line 123-124. Anthelmintic treatment during dry periods, meanwhile, is likely to be highly selective for resistance due to lack of pasture refugia.   Do you mean parasite resistance to AH?

Line 125-126. areas experiencing more arid conditions under climate change might consequently suffer accelerating drug resistance. The parasite burden in arid zones is lower than tropical conditions. So, you need to explain this aspect.

Line 164-165. Furthermore, the CC has a negative impact on the food supply for the human population. A good example of this phenomena are the rural communities from the wetlands in Mexico. This paragraph needs edition and a couple of lines explaining that Mexican families of the countryside use wild mammals as food resources.

Line 170-174. Climate change is affecting the worldwide vegetation, and Mexico is not the exception, another critical factor is also the deforestation of the different types of forest, particularly in the tropical moist forest ecoregions [37]. Delete this paragraph because it does not have a relationship with the topic or include how deforestation influences the habitat of wild animals.

Line 181-185. An incredible effort is required to understand how to climate change could affect wildlife communities in Mexico. Therefore, it is a big chance to researches to design studies which involve a multidisciplinary approach (climatic conditions, nutrition, reproduction, survivorship, etc.). Furthermore, that point of view will allow us to have an overall picture of the current and most important further situation of wild animal communities.

Need edition. Ex, “An effort is required to know how to climate change affect wildlife communities in Mexico. Therefore, the researchers have the chance to design studies involving a multidisciplinary approach (climatic conditions, nutrition, reproduction, survivorship). Furthermore, that point of view will allow us to have an overall picture of the current and further situation of wild animal communities”.

Line 212. after being run over and killed in. Would you please change the verb killed by Sacrificed.

Line 212-213. Subsequently, an analysis of the parasites that could be found in each organ was carried out in the laboratory. Do you mean different organs were dissected to obtain parasites?

223-224. This order is important because humans are classified in this order. Delete this line.

Line 297-301. Climate change as described in other sections is the modification of the atmosphere, climatic seasons, solar radiation, temperatures, gases in the atmosphere attributed to natural variability or human activities [74]. These changes in the environment modify the climate in unforeseen ways. In addition, they have had a general effect on global temperatures, rainfall, among others. Therefore, in general, living organisms seek ways to adapt to climate change. Would you please delete this paragraph because it was mentioned previously?

Line 315-316. Several authors agree that the highest numbers of parasites occur in the post-rainy season. Need cites.

Line 485-538. This text is about techniques and methods to identify animals resistant to GIN infections. So, this text could be deleted.

Line 562-563.  Conclusion 2. New alternatives for controlling gastrointestinal parasitic nematode infections of ruminants are required. The authors can not conclude about the alternatives to control GIN infections because they just showed information about animal genetic resistance.

Author Response

Reviewer 2

The writing presents a topic of interest that impacts animal production and the economy of producers. State of the art on the interaction between animals and parasitism under the environmental challenge of climate change is essential for science and humanity. The structure of the review is interesting and raises the current situation of the actors and their future. It shows the areas of opportunity that must be addressed to decrease the negative impact of climate change on animal welfare and animal production. It proposes the selection of animals where production and resistance to pathogens are integrated. This document will be of interest to scientists and producers of Small ruminants around the world.

Authors answers: Comments are greatly appreciated.

The title mentions that the topic is about ruminants. However, more than 90% of the text refers to Small ruminants (sheep and goats). Therefore I suggest that this point must be adjusted in the title. If the authors prefer to keep Ruminants, then they must include more information about GIN infection in cattle. But ticks are more critical than GIN in this animal. A Review of the Impact of Climate Change on the Epidemiology of Gastrointestinal Nematode Infections in Small Ruminants and Wildlife.

Authors answers: The correction was made. Page 1.

Line 33. Could you explain “el niño” phenomenon”?

Authors answers: The correction was made. Page 1.

Line 94. Please upload to line 93

Authors answers: The correction was made. Page 3.

Line 116-119. This paragraph needs an edition. Temperature and rainfall patterns influence parasite transmission. However, those factors are directly related to parasite burden.

Authors answers: The correction was made. Page 3.

Line 123-124. Anthelmintic treatment during dry periods, meanwhile, is likely to be highly selective for resistance due to lack of pasture refugia. Do you mean parasite resistance to AH?

Authors answers: The correction was made. Page 4.

Line 125-126. areas experiencing more arid conditions under climate change might consequently suffer accelerating drug resistance. The parasite burden in arid zones is lower than tropical conditions. So, you need to explain this aspect.

Authors answers: The correction was made. Page 4.

Line 164-165. Furthermore, the CC has a negative impact on the food supply for the human population. A good example of this phenomena are the rural communities from the wetlands in Mexico. This paragraph needs edition and a couple of lines explaining that Mexican families of the countryside use wild mammals as food resources.

Authors answers: The correction was made. Page 4.

Line 170-174. Climate change is affecting the worldwide vegetation, and Mexico is not the exception, another critical factor is also the deforestation of the different types of forest, particularly in the tropical moist forest ecoregions [37]. Delete this paragraph because it does not have a relationship with the topic or include how deforestation influences the habitat of wild animals.

Authors answers: The correction was made. Page 4-5.

Line 181-185. An incredible effort is required to understand how to climate change could affect wildlife communities in Mexico. Therefore, it is a big chance to researches to design studies which involve a multidisciplinary approach (climatic conditions, nutrition, reproduction, survivorship, etc.). Furthermore, that point of view will allow us to have an overall picture of the current and most important further situation of wild animal communities.

Authors answers: The correction was made. Page 5.

Need edition. Ex, “An effort is required to know how to climate change affect wildlife communities in Mexico. Therefore, the researchers have the chance to design studies involving a multidisciplinary approach (climatic conditions, nutrition, reproduction, survivorship). Furthermore, that point of view will allow us to have an overall picture of the current and further situation of wild animal communities”.

Authors answers: The correction was made. Page 4-5.

Line 212. after being run over and killed in. Would you please change the verb killed by Sacrificed.

Authors answers: The correction was made. Page 5.

Line 212-213. Subsequently, an analysis of the parasites that could be found in each organ was carried out in the laboratory. Do you mean different organs were dissected to obtain parasites?

Authors answers: The correction was made. Page 5.

223-224. This order is important because humans are classified in this order. Delete this line.

Authors answers: The correction was made. Page 5.

Line 297-301. Climate change as described in other sections is the modification of the atmosphere, climatic seasons, solar radiation, temperatures, gases in the atmosphere attributed to natural variability or human activities [74]. These changes in the environment modify the climate in unforeseen ways. In addition, they have had a general effect on global temperatures, rainfall, among others. Therefore, in general, living organisms seek ways to adapt to climate change. Would you please delete this paragraph because it was mentioned previously?

Authors answers: The correction was made. Page 7.

Line 315-316. Several authors agree that the highest numbers of parasites occur in the post-rainy season. Need cites.

Authors answers: The correction was made. Page 3.

Line 485-538. This text is about techniques and methods to identify animals resistant to GIN infections. So, this text could be deleted.

Authors answers: The correction was made. Page 10 and11.

Line 562-563.  Conclusion 2. New alternatives for controlling gastrointestinal parasitic nematode infections of ruminants are required. The authors can not conclude about the alternatives to control GIN infections because they just showed information about animal genetic resistance.

Authors answers: The correction was made. Page 11.

Reviewer 3 Report

Dear Authors,

I have a few changes suggestion listed below:

I suggest to chane title to "A review of the impact of climate chage on the epizootiology of gatsroitestinal nematode infections in wildlife and ruminats" in relation to order of descibed issues. Therm epidemiology should be used in the context to diseases of humans, epizootiology is the eqivalent used for diseases of animals (see also: page 3, line 105 and p. 8 , l. 345.

Page 2, line 49 - Latin species names (e.g Trichostrongylus) need to be presented in full form, need "sp." after genus name or should be preceded by "genus/genera". In page 5, line 216-217 - I have the same sugestion.

Page 6, line 257, 266 and page 7, line 284, 290 - citations in text should have autors surname (e.g. "Kaplan [63] also mentions...")

Page 7, line 278 - I suggest use words "fecal pellets" instead "fecal cake"

Page 6, line 269 - I suggest use term "infective stage" intead "pre-parasitic forms"

Author Response

Reviewer 3

I have a few changes suggestion listed below:

I suggest to chane title to "A review of the impact of climate chage on the epizootiology of gatsroitestinal nematode infections in wildlife and ruminats" in relation to order of descibed issues.

Authors answers: The correction was made.

Therm epidemiology should be used in the context to diseases of humans, epizootiology is the eqivalent used for diseases of animals (see also: page 3, line 105 and p. 8 , l. 345.

Authors answers: The correction was made in the title and on line 345 of the manuscript.

Page 2, line 49 - Latin species names (e.g Trichostrongylus) need to be presented in full form, need "sp." after genus name or should be preceded by "genus/genera". In page 5, line 216-217 - I have the same sugestion.

Authors answers: The correction was made.

Page 6, line 257, 266 and page 7, line 284, 290 - citations in text should have autors surname (e.g. "Kaplan [63] also mentions...")

Authors answers: The correction was made.

Page 7, line 278 - I suggest use words "fecal pellets" instead "fecal cake"

Authors answers: The correction was made.

Page 6, line 269 - I suggest use term "infective stage" intead "pre-parasitic forms"

Authors answers: The correction was made.

Reviewer 4 Report

Review Article: “A Review of the Impact of Climate Change on the Epidemiology of Gastrointestinal Nematode Infections in Ruminants and Wildlife”, submitted to Pathogens

The manuscript reviews several issues regarding the impact of climate change (CC) on animal hosts (livestock and wildlife) and parasite epidemiology, and selection of resistant hosts. The understanding of the impacts derived from climate change on parasite infection dynamics and livestock husbandry is an urgent challenge, and therefore this effort is timely and within the scope of Pathogens. 

However, the current manuscript is mostly an uncritical compilation of selected previous research, with an incoherent text in several sections and a very confusing structure and presentation of the several topics that aims to cover. The impacts of CC on the world’s livestock and parasite populations are very complex and non-linear, as the authors recognise in several sections of the manuscript (Lines 106-114, 132, 148, etc.). In this regard, it is not clear what are the exact aims and research questions that drive this review, and these are particularly missing at the Introduction section. In other words: What is/are the driving question/s that the authors wish to answer with this review? And how the sections of the manuscript do respond to those driving questions? Moreover, it is not clear how the reviewed articles were curated and analysed based on the aims/driving research questions.

The content and presentation of the current manuscript is confusing, moving without a clear structure between different topics and regions of the world (from general effects of CC on animals in a global scale to local effects in Mexico, and then back to a global scale), and without a clear guiding research question (see comment above). The manuscript starts listing effects of CC on livestock (mainly focused on small ruminants and heat waves), followed by collecting information of the indirect effects of livestock systems on parasite epidemiology (mainly on cattle – based on the references). Then the manuscript moves into describing the impacts of CC on wild mammals in Mexico (including impacts on food production and forests?), following by a confusing list of diverse parasite species (not only nematodes, as the title of the article would suggest) reported in some mammals in Mexico and Latin America, without a clear connection with the previous sections. Then the article moves to describe the effects of GI nematodes on small ruminants and cattle (with focus on  H. contortus), based on a compilation of results from previous articles and a very incoherent and erratic text, while describing well-known facts of the impact of climate (e.g. rainfall, moisture, etc.) on free-living stages of GI nematodes. The article then describes the need to develop adaptation and mitigation strategies in livestock to cope with diverse challenges (including CC), repeating some impacts of CC on livestock already mentioned above, then listing some of these adaptation strategies (including selection of resilient breeds and nutrition), but then moving into effects of heat stress on animal and productive characteristics with apparently no clear connection with the impact of heat stress on the response of the host to parasitic nematodes. Finally, the article emphasises the relevance of selecting parasite-resistant and resilient host breeds, including a very thorough description of current genomic research to identify candidate genes related with parasite resistance in small ruminants, which is perhaps the best section in the manuscript. However, the contribution of the manuscript as a whole is not clear.

A useful review article is not merely a collection of previous references, but a critical analysis of the published literature on a particular research topic guided by clear scientific questions that can help the readers to navigate in the available (and always increasing) body of knowledge, aiming to shed light and to suggest further research efforts to advance the knowledge on that topic. Consequently, it is difficult to evaluate the scientific quality and contribution of this review in its current version. In this regard, the manuscript cannot be accepted in its current version and should be completely re-structured. The following general comments are given as suggestions for the preparation of a new submission: 

- Re-structure the manuscript with the aim to prepare a coherent article, with clear connections between its sections. 

- Define the key research questions that the manuscript should attempt to respond (see comment above). In my opinion, the manuscript should focus on defining and answering clear questions regarding the impact of CC on small ruminants and their parasites (perhaps in defined climatic/production settings, e.g. tropical/subtropical), and which strategies exist and should be further developed to enhance the adaptive capacity of small ruminants to cope with parasite infections in a changing climate.

- I would suggest removing the current sections on wildlife (Sections 4. and 5.). If the authors wish to keep the wildlife sections, please provide an argument of why this should be kept. 

- Merge Sections 6 and 7, and re-write this section with a coherent text and a clear summary of what are the main impacts of CC on parasitic nematodes of small ruminants/H. contortus (perhaps with focus on tropical/subtropical regions).

- Merge Sections 8 and 9, with focus on the development of resilient breeds with enhance adaptability to cope with parasite infections in a changing climate and the genomic research to identify genes linked with host resistant, while avoiding the unnecessary listing of references on topics dealing with productive characteristics not directly linked with resilience to GI nematode pathogens. 

- Add a new section proposing further research questions that need to be answered in order to enhance the adaptability of small ruminants to parasite infections in a changing climate, including challenges and potential of selection of resistant breeds and other adaptation/mitigation strategies. 

Author Response

Reviewer 4

Comments and Suggestions for Authors

Review Article: “A Review of the Impact of Climate Change on the Epidemiology of Gastrointestinal Nematode Infections in Ruminants and Wildlife”, submitted to Pathogens.

The manuscript reviews several issues regarding the impact of climate change (CC) on animal hosts (livestock and wildlife) and parasite epidemiology, and selection of resistant hosts. The understanding of the impacts derived from climate change on parasite infection dynamics and livestock husbandry is an urgent challenge, and therefore this effort is timely and within the scope of Pathogens. 

Authors answers: Comments are greatly appreciated.

However, the current manuscript is mostly an uncritical compilation of selected previous research, with an incoherent text in several sections and a very confusing structure and presentation of the several topics that aims to cover. The impacts of CC on the world’s livestock and parasite populations are very complex and non-linear, as the authors recognise in several sections of the manuscript (Lines 106-114, 132, 148, etc.). In this regard, it is not clear what are the exact aims and research questions that drive this review, and these are particularly missing at the Introduction section. In other words: What is/are the driving question/s that the authors wish to answer with this review? And how the sections of the manuscript do respond to those driving questions? Moreover, it is not clear how the reviewed articles were curated and analysed based on the aims/driving research questions.

Authors answers: The correction was made. Page 3 and 4.

The content and presentation of the current manuscript is confusing, moving without a clear structure between different topics and regions of the world (from general effects of CC on animals in a global scale to local effects in Mexico, and then back to a global scale), and without a clear guiding research question (see comment above). The manuscript starts listing effects of CC on livestock (mainly focused on small ruminants and heat waves), followed by collecting information of the indirect effects of livestock systems on parasite epidemiology (mainly on cattle – based on the references). Then the manuscript moves into describing the impacts of CC on wild mammals in Mexico (including impacts on food production and forests?), following by a confusing list of diverse parasite species (not only nematodes, as the title of the article would suggest) reported in some mammals in Mexico and Latin America, without a clear connection with the previous sections. Then the article moves to describe the effects of GI nematodes on small ruminants and cattle (with focus on  H. contortus), based on a compilation of results from previous articles and a very incoherent and erratic text, while describing well-known facts of the impact of climate (e.g. rainfall, moisture, etc.) on free-living stages of GI nematodes. The article then describes the need to develop adaptation and mitigation strategies in livestock to cope with diverse challenges (including CC), repeating some impacts of CC on livestock already mentioned above, then listing some of these adaptation strategies (including selection of resilient breeds and nutrition), but then moving into effects of heat stress on animal and productive characteristics with apparently no clear connection with the impact of heat stress on the response of the host to parasitic nematodes.

Authors answers: The correction was made.

Finally, the article emphasises the relevance of selecting parasite-resistant and resilient host breeds, including a very thorough description of current genomic research to identify candidate genes related with parasite resistance in small ruminants, which is perhaps the best section in the manuscript.

Authors answers: Comments are greatly appreciated.

However, the contribution of the manuscript as a whole is not clear.

Authors answers: Comments are greatly appreciated. The correction was made in all manuscript.

A useful review article is not merely a collection of previous references, but a critical analysis of the published literature on a particular research topic guided by clear scientific questions that can help the readers to navigate in the available (and always increasing) body of knowledge, aiming to shed light and to suggest further research efforts to advance the knowledge on that topic. Consequently, it is difficult to evaluate the scientific quality and contribution of this review in its current version. In this regard, the manuscript cannot be accepted in its current version and should be completely re-structured. The following general comments are given as suggestions for the preparation of a new submission: 

- Re-structure the manuscript with the aim to prepare a coherent article, with clear connections between its sections. 

Authors answers: Comments are greatly appreciated. The correction was made in all manuscript.

- Define the key research questions that the manuscript should attempt to respond (see comment above). In my opinion, the manuscript should focus on defining and answering clear questions regarding the impact of CC on small ruminants and their parasites (perhaps in defined climatic/production settings, e.g. tropical/subtropical), and which strategies exist and should be further developed to enhance the adaptive capacity of small ruminants to cope with parasite infections in a changing climate.

Authors answers: We included the objective of the review in the abstract and introduction section.

- I would suggest removing the current sections on wildlife (Sections 4. and 5.). If the authors wish to keep the wildlife sections, please provide an argument of why this should be kept. 

Authors answers: The correction was made.

- Merge Sections 6 and 7, and re-write this section with a coherent text and a clear summary of what are the main impacts of CC on parasitic nematodes of small ruminants/H. contortus (perhaps with focus on tropical/subtropical regions).

Authors answers: Those sections were merged as requested.

- Merge Sections 8 and 9, with focus on the development of resilient breeds with enhance adaptability to cope with parasite infections in a changing climate and the genomic research to identify genes linked with host resistant, while avoiding the unnecessary listing of references on topics dealing with productive characteristics not directly linked with resilience to GI nematode pathogens. 

Authors answers: The fusion was carried out.

- Add a new section proposing further research questions that need to be answered in order to enhance the adaptability of small ruminants to parasite infections in a changing climate, including challenges and potential of selection of resistant breeds and other adaptation/mitigation strategies. 

Authors answers: In section 4 the information was added.

Round 2

Reviewer 2 Report

Dear editor, after reviewing the manuscript and confirming the changes suggested to the present, there are two minor corrections that must be adressed.

1.- Line  458-459. From these measures, FAMACHA score has been widely utilized by producers for assessment of parasite resistance and has facilitated accumulation of records [115]. Mandonnet, N., Menendez-Buxadera, A., Arquet, R., Mahieu, M., Bachand, M., Aumont, G. Genetic variability in resistance to gastro-intestinal strongyles during early lactation in Creole goats. Animal Science. 2007, doi.org/ 10.1079/ASC200640.

Those authors worked with goats and measured FEC and PCV, but not FAMACHA. I agree about the high correlation between FAMACHA and PCV. So, you should edit the paragraph.

2.- Line 463. Severe internal parasite infestation. It must be changed by "Severe internal parasite infection".

Sincerely

Author Response

Reviewer 2

1. Line 458-459. From these measures, FAMACHA score has been widely utilized by producers for assessment of parasite resistance and has facilitated accumulation of records [115]. Mandonnet, N., Menendez-Buxadera, A., Arquet, R., Mahieu, M., Bachand, M., Aumont, G. Genetic variability in resistance to gastro-intestinal strongyles during early lactation in Creole goats. Animal Science. 2007, doi.org/ 10.1079/ASC200640. Those authors worked with goats and measured FEC and PCV, but not FAMACHA. I agree about the high correlation between FAMACHA and PCV. So, you should edit the paragraph.
Authors answers: The change was made.

3. Line 463. Severe internal parasite infestation. It must be changed by "Severe internal parasite infection".
Authors answers: The change was made.

Reviewer 4 Report

The new revised version is an improvement from the previous manuscript. However, it stills need further modifications. My major comments are:

- Title: Based on the new objective of the revised manuscript, and the comments presented below, I would suggest modifying the title to “A review of the impact of climate change on small ruminant systems and gastrointestinal nematodes in tropical conditions”

- I previously suggested the reviewers to remove the section on wildlife (now section "3. Influence of Climate Change in Wild Mammals in Mexico"), or otherwise provide an argument of why this section is relevant for this manuscript. However, the authors have yet to respond to this point. I suggest once again to remove the whole section 3. on wildlife from the text and from the title, and just focus the manuscript on the impacts of CC on small ruminant systems and their GI parasites under tropical conditions.

- I also previously suggested to add a text proposing further research questions to guide future studies aiming to enhance the adaptability of small ruminants to parasite infections in a changing climate. To this, the authors have responded that this information was added in Section 4. However, I cannot find it. Perhaps the authors can present this information as a brief section before the Conclusions.

- L105-109 in Section 2 should be moved to Section “Effect of Climate Change on Nematode Parasites of Livestock”.

- L238-239: Please re-write this sentence. What do the authors mean with “we mention the refugia phenomenon”? Perhaps provide a brief explanation of the concept of refugia with a reference (could be the same reference 31 Hodgkinson et al 2019, IJPDDR).

- L246-248: Please re-write this sentence.  

- L256-258: Delete this sentence, it is unconnected with the rest of the section and refers to a study on cattle in Northern North America. This manuscript is focused on small ruminants in tropical conditions.

-L263: Replace “cybals” by “faeces”.

-L288-292: Please provide here a concrete definition of resilience in livestock and the individual animal, and describe some examples of features associated with host resilience against parasite infections (e.g. sustained production levels despite parasite infections, etc.). Perhaps place here the definition from L429

-L293-298: Please explain the concept of adaptability. Do the authors use adaptability as a synonym of resilience? If this is the case, I suggest replacing animal adaptability for animal resilience for clarity.

-L299: What do the authors mean with “mitigation needs of livestock”? Is it mitigation of livestock systems to climate change effects? Please clarify.

-L313-314: Please re-write this sentence.

-L316-324: This text is a repetition of issues already presented. I suggest deleting it.

-L412: Please rephrase the title of this section to reflect better its content. A suggestion could be: “Selection for parasite resistance in small ruminants under climate change”

- L414: Please add here at the beginning of the section a sentence on how selection of animals for parasite resistance could be a sustainable response to higher parasite risk due to climate change (to try to link this last section with the whole manuscript).

-L522-528: Here in the conclusion, please respond briefly to the objective of the manuscript: How climate change has impacted the small ruminants and the epidemiology of their GI nematodes in tropical areas? And please remove “wild animals” (see comment above).

Author Response

Reviewer 4
The new revised version is an improvement from the previous manuscript. However, it stills need further modifications. My major comments are:
- Title: Based on the new objective of the revised manuscript, and the comments presented below, I would suggest modifying the title to “A review of the impact of climate change on small ruminant systems and gastrointestinal nematodes in tropical conditions”
Authors answers: The title was changed “A Review of the Impact of Climate Change on the Epidemiology of Gastrointestinal Nematode Infections in Small Ruminants and Wildlife in tropical conditions”.

I previously suggested the reviewers to remove the section on wildlife (now section "3. Influence of Climate Change in Wild Mammals in Mexico"), or otherwise provide an argument of why this section is relevant for this manuscript. However, the authors have yet to respond to this point. I suggest once again to remove the whole section 3. on wildlife from the text and from the title, and just focus the manuscript on the impacts of CC on small ruminant systems and their GI parasites under tropical conditions.
Authors answers: The section "Influence of Climate Change in Wild Mammals in Mexico" will not be deleted from the writing. According to the title, this information is relevant and very important, because wild animals are one of the most affected by climate change. Sometimes these species and orders are neglected, but they are of great importance, and we also believe that it is an unpublished compilation of these subjects.

I also previously suggested to add a text proposing further research questions to guide future studies aiming to enhance the adaptability of small ruminants to parasite infections in a changing climate. To this, the authors have responded that this information was added in Section 4. However, I cannot find it. Perhaps the authors can present this information as a brief section before the Conclusions.
Authors answers: It was added to the section entitled: “6 Future Studies Aiming to Enhance the Adaptability of Small Ruminants to Parasite Infections in a Changing Climate” as indicated by the reviewer.

L105-109 in Section 2 should be moved to Section “Effect of Climate Change on Nematode Parasites of Livestock”.
Authors answers: The change was made in Section 2 should be moved to Section “Effect of Climate Change on Nematode Parasites of Livestock”.

L238-239: Please re-write this sentence. What do the authors mean with “we mention the refugia phenomenon”? Perhaps provide a brief explanation of the concept of refugia with a reference (could be the same reference 31 Hodgkinson et al 2019, IJPDDR).
Authors answers: The information was added

L246-248: Please re-write this sentence.
Authors answers: The sentence was rewritten.
“The infecting larvae (L3) of the GIN need humidity to survive in the grasslands; However, when the rains are scarce, the populations of the GIN decrease due to this situation”.

L256-258: Delete this sentence, it is unconnected with the rest of the section and refers to a study on cattle in Northern North America. This manuscript is focused on small ruminants in tropical conditions.
Authors answers: Sentence was deleted.

-L263: Replace “cybals” by “faeces”.
Authors answers: The change was made.

-L288-292: Please provide here a concrete definition of resilience in livestock and the individual animal, and describe some examples of features associated with host resilience against parasite infections (e.g. sustained production levels despite parasite infections, etc.). Perhaps place here the definition from L429
Authors answers: The definition was added and some examples of resilience in animals were also added.

-L293-298: Please explain the concept of adaptability. Do the authors use adaptability as a synonym of resilience? If this is the case, I suggest replacing animal adaptability for animal resilience for clarity.
Authors answers: The reviewer is right, the change from “adaptability” to “resilience” was made.

-L299: What do the authors mean with “mitigation needs of livestock”? Is it mitigation of livestock systems to climate change effects? Please clarify.
Authors answers: The sentence was modified.

-L313-314: Please re-write this sentence.
Authors answers: The sentence was eliminated.

-L316-324: This text is a repetition of issues already presented. I suggest deleting it.
Authors answers: The change was made.

-L412: Please rephrase the title of this section to reflect better its content. A suggestion could be: “Selection for parasite resistance in small ruminants under climate change”
Authors answers: The change was made.

L414: Please add here at the beginning of the section a sentence on how selection of animals for parasite resistance could be a sustainable response to higher parasite risk due to climate change (to try to link this last section with the whole manuscript).
Authors answers: The sentence was modified

-L522-528: Here in the conclusion, please respond briefly to the objective of the manuscript: How climate change has impacted the small ruminants and the epidemiology of their GI nematodes in tropical areas? And please remove “wild animals” (see comment above).
Authors answers: The discussion section was modified based on the reviewer's valuable comment